# A Candidate RNAi Screen Reveals Diverse RNA-Binding Protein Phenotypes in *Drosophila* Flight Muscle

**DOI:** 10.3390/cells10102505

**Published:** 2021-09-22

**Authors:** Shao-Yen Kao, Elena Nikonova, Sabrina Chaabane, Albiona Sabani, Alexandra Martitz, Anja Wittner, Jakob Heemken, Tobias Straub, Maria L. Spletter

**Affiliations:** 1Biomedical Center, Department of Physiological Chemistry, Ludwig-Maximilians-Universität München, Großhaderner Str. 9, 82152 Martinsried-Planegg, Germany; shao-yen.kao@bmc.med.lmu.de (S.-Y.K.); elena.nikonova@bmc.med.lmu.de (E.N.); sabrina.chaabane@tum.de (S.C.); wittner@genzentrum.lmu.de (A.W.); Jakob.Heemken@campus.lmu.de (J.H.); 2Department of Biology, University of Wisconsin at Madison, 1117 W. Johnson St., Madison, WI 53706, USA; sabani2@wisc.edu; 3Molecular Nutrition Medicine, Else Kröner-Fresenius Center, Technical University of Munich, 85354 Freising, Germany; alexandra.martitz@ndcls.ox.ac.uk; 4Biomedical Center, Bioinformatics Core Facility, Ludwig-Maximilians-Universität München, Großhaderner Str. 9, 82152 Martinsried-Planegg, Germany; tstraub@bmc.med.lmu.de

**Keywords:** RNA binding proteins, SF1, Hrb87F, Bru1, *Drosophila*, flight muscle, RNAi, splicing

## Abstract

The proper regulation of RNA processing is critical for muscle development and the fine-tuning of contractile ability among muscle fiber-types. RNA binding proteins (RBPs) regulate the diverse steps in RNA processing, including alternative splicing, which generates fiber-type specific isoforms of structural proteins that confer contractile sarcomeres with distinct biomechanical properties. Alternative splicing is disrupted in muscle diseases such as myotonic dystrophy and dilated cardiomyopathy and is altered after intense exercise as well as with aging. It is therefore important to understand splicing and RBP function, but currently, only a small fraction of the hundreds of annotated RBPs expressed in muscle have been characterized. Here, we demonstrate the utility of *Drosophila* as a genetic model system to investigate basic developmental mechanisms of RBP function in myogenesis. We find that RBPs exhibit dynamic temporal and fiber-type specific expression patterns in mRNA-Seq data and display muscle-specific phenotypes. We performed knockdown with 105 RNAi hairpins targeting 35 RBPs and report associated lethality, flight, myofiber and sarcomere defects, including flight muscle phenotypes for *Doa*, *Rm62*, *mub*, *mbl*, *sbr*, and *clu*. Knockdown phenotypes of spliceosome components, as highlighted by phenotypes for A-complex components SF1 and Hrb87F (hnRNPA1), revealed level- and temporal-dependent myofibril defects. We further show that splicing mediated by SF1 and Hrb87F is necessary for Z-disc stability and proper myofibril development, and strong knockdown of either gene results in impaired localization of kettin to the Z-disc. Our results expand the number of RBPs with a described phenotype in muscle and underscore the diversity in myofibril and transcriptomic phenotypes associated with splicing defects. *Drosophila* is thus a powerful model to gain disease-relevant insight into cellular and molecular phenotypes observed when expression levels of splicing factors, spliceosome components and splicing dynamics are altered.

## 1. Introduction

Animals, from flies to humans, possess hundreds of distinct muscles that are characterized by differences in both morphology and contractile properties [1]. An outstanding question in developmental biology is how this diversity is generated, and in particular, which cytoskeletal adaptations support such a diverse range of contractile abilities. Sarcomeres, the simplest contractile subunits of muscle, are mini-motors driven by the interaction between antiparallel filaments of actin, anchored at the Z-disc, and myosin, anchored at the M-line [2]. Sarcomeres are built of more than one hundred proteins and assemble end-to-end into long myofibrils that span the length of a muscle [3]. Small changes in the structure and ratios of these sarcomeric proteins through altered gene expression, alternative splicing, or messenger ribonucleic acid (mRNA) regulatory dynamics can alter the biomechanics of muscle contraction, and thereby serve as a mechanism to fine-tune contractile properties [4,5,6,7]. For example, expression of short, stiff Titin isoforms in the heart and longer, more flexible Titin isoforms in skeletal muscle contribute to physiological differences among these muscle types [8,9]. Misregulation of ribonucleic acid (RNA) processing therefore leads to a vast collection of muscle diseases from myotonic dystrophies to cardiomyopathies [10,11,12,13]. Moreover, exercise, adaptation, aging and recovery after injury also involve changes in RNA regulation [14,15,16]. Thus, understanding normal RNA regulatory dynamics in developing muscle is foundational to understanding normal muscle physiology as well as disease.

### 1.1. RNA-Binding Proteins with Diverse Structures and Functions Regulate RNA Dynamics

RNA binding proteins (RBPs), which recognize and bind defined RNA sequences, control all steps in the RNA life-cycle, from exon definition and alternative splicing to mRNA trafficking, nuclear export, localization, translation dynamics and turn-over [17,18]. RBPs therefore play a key role in determining which structural gene isoforms are expressed in muscle, in which ratios and with which temporal and fiber-type specific dynamics [7,11]. Canonical RBPs have one or more RNA-binding domains, for example an RNA-recognition motif (RRM), hnRNPK homology domain (KH), double-stranded RNA-binding domain (dsRBD) or DEAD/DEAH box [4]. Other RBPs bind RNA through intrinsically disordered, or unstructured regions. RNA interactome capture (RIC) studies, which unbiasedly identify RNA-bound proteins, suggest that hundreds of such proteins lacking canonical RNA-binding domains are capable of binding RNA in a developmentally or spatially regulated manner [19,20]. RBPs often contain additional protein domains, from Zinc fingers and PDZ (PSD-95/Discs-large/ZO1) domains that facilitate protein–protein interactions and the assembly of multimeric RBP regulatory complexes such as LASR (large assembly of splicing regulators) [21] to G-patch and serine-rich (RS) domains frequently associated with splicing factors [22] to prion-like domains (PrLDs) that together with RNA binding facilitate the formation of phase-separated liquid compartments [23]. This illustrates that a major challenge in deciphering RNA regulatory dynamics in healthy and diseased muscle is the tissue-type specific identification of RBPs and their functions.

### 1.2. Regulation of RBPs Modulates RNA Processing in Muscle

Functionally, RBPs often have manifold RNA targets and can regulate multiple steps in RNA processing. A survey of 56 RBPs in the immortalized Schneider 2 (S2) cell culture line derived from late-stage *Drosophila* embryos found that individual RBPs can regulate from tens to hundreds of splice events [24], while studies in vertebrates suggest that CELF (CUG-BP and Etr-3-like factor), RBFOX (RNA binding Fox-1 homolog) and MBNL (muscleblind-like) family proteins may regulate hundreds to thousands of events in muscle [25,26]. TDP-43 (TAR DNA-binding protein 43), which is associated with neurodegenerative disorders, inclusion body myositis and rimmed vacuole myopathies, regulates alternative splicing as well as mRNA trafficking and translation and mRNP granule formation [27,28]. CELF and MBNL proteins, which have well-characterized roles during muscle development, regulate mRNA stability, translation and localization in addition to alternative splicing [26,29,30]. These RBPs further function as part of regulatory networks subject to cooperativity, feedback and cross-regulation, as exemplified in muscle by interactions among RBFOX, CELF and MBNL family members [11,25,31,32]. RBPs themselves are also subject to multiple levels of regulation. For example, RBP activity or expression level can be temporally regulated: higher levels of CELF1 expression during early myogenesis promote embryonic splicing, while increased levels of MBNL1 at later stages of differentiation promote post-natal splicing patterns in skeletal muscle [33]. RBPs can be alternatively spliced to produce different isoforms that localize to distinct sub-cellular compartments, thus executing different functions: nuclear-localized MBNL isoforms regulate alternative splicing, while cytoplasmic-localized MBNL isoforms regulate mRNA trafficking and stability. RBPs can be post-translationally modified to regulate their activity and stability: Stk38 (serine/threonine kinase 38) binds and modifies Rbm24 (RNA binding motif protein 24) in cardiomyocytes, stabilizing the protein and promoting Rbm24 mediated splicing of structural proteins necessary for sarcomere assembly [34]. RBPs are thus central to defining RNA regulatory dynamics in muscle, both through their regulation of target mRNAs as well as through regulation of their own expression, localization and activity.

### 1.3. Drosophila as a Model to Identify and Study Muscle-Specific RBP Function

Despite the hundreds of proteins identified to bind RNA or contain canonical RNA-binding domains [20,35,36], our understanding of RNA regulatory dynamics is still vastly incomplete as relatively few RBPs have been studied extensively in muscle. Most well-studied RBPs in muscle are associated with or causal for muscle diseases, notably CELF1 and MBNL1 (myotonic dystrophy) [37], SMN (spinal muscular atrophy) [38], TDP-43 [28], and Rbm20 and Rbm24 (dilated cardiomyopathy) [39]. We recently found that only 82 RBPs have been studied in muscle in any model organism, which is far less than the hundreds of annotated RBPs expressed in transcriptomic data from muscle [11]. Even well-characterized RBPs are often incompletely understood because of tissue-specific regulation, a lack of attention to RBP isoform-specific function and pleotropic function at multiple levels in the RNA regulatory hierarchy [40].

To address this need to both identify new RBPs and characterize mechanisms of RBP function and regulation in muscle, we present the *Drosophila* adult musculature as an attractive genetic model system. The adult musculature in *Drosophila* displays functional diversity and developmental plasticity similar to vertebrate muscles [15]. Structural components of the sarcomere and many aspects of muscle physiology are highly conserved [2,41]. Disease models in the fly have proven highly informative for many genes implicated in human muscle disease, for example *CELF1*, *MBNL*1, *SMN* (Survival motor neuron) and *DMD* (Dystrophin) [42,43,44]. Flies have a particularly diverse and powerful genetic toolbox that can be employed in vivo, including well-established tools to regulate spatial and temporal expression and to fine-tune levels of both RNAi-mediated knockdown and transgene expression [45,46,47]. These characteristics make the fly an attractive model to both screen for RBPs with muscle phenotypes as well as to investigate conserved molecular mechanisms of RBP function.

### 1.4. Fiber-Types and Development of the Drosophila Adult Musculature

The adult musculature of *Drosophila* is built in under 100 h during pupal development [43,48]. Myogenic precursors that were set aside during embryogenesis proliferate associated with imaginal discs to generate thousands of myoblasts [49,50,51]. During metamorphosis, the myoblasts migrate to sites of muscle formation and fuse with founder cells that specify the identity of the forming myotubes [52,53,54]. One exception are the dorsal–longitudinal indirect flight muscles (DLMs), which form via myoblast fusion to three larval template muscles that escape histolysis [42,43]. The newly formed myotubes then grow and actively search for their target tendon cells [48,55]. After tendon attachment, sarcomeres and myofibrils self-assemble in the developing myofibers in a tension-dependent manner [2,56]. Spontaneous contractions during subsequent development help organize and refine sarcomere structure, and a coordinated shift in transcription promotes myofiber maturation [57,58,59,60]. This general process occurs in all muscles, but differential gene and splice isoform expression lead to distinct morphologies and biophysical properties among muscle fiber-types [6,7,61,62].

Adult *Drosophila* possess more than 150 distinct muscle groups that are characterized into two morphologically and functionally distinct fiber types: tubular and fibrillar [48,63]. The majority of body muscles are tubular, characterized by fused and laterally aligned myofibrils surrounding a central core of nuclei [6,48]. Tubular muscles are synchronous, meaning that contraction is coupled to neuronal impulses [64,65,66]. They control fine movements such as mating, pupal eclosion, wing positioning (controlled by the direct flight muscles, DFM), ambulation (controlled by leg muscles) and jumping (controlled by the tergal depressor of the trochanter, TDT) [2,6,67]. It is likely that different tubular fiber-types exist based on differential gene expression and splicing patterns [6,68,69,70], but they have not yet been systematically defined and characterized. In contrast to tubular muscle, myofibrils in the fibrillar indirect flight muscles (IFMs) do not fuse or laterally align, in part due to their close interactions with mitochondria [71,72], and nuclei are distributed throughout the IFM myofiber [73,74]. The IFM are stretch activated and asynchronous, meaning that contraction is enhanced by stretch and decoupled from neuronal impulses, an adaptation that allows IFM to contract at rates up to 200 hertz and generate power up to 80 Watts per kg [66,67]. IFM consists of two antiparallel sets of six dorsal longitudinal muscles (DLM), which are responsible for the wing downstroke, and seven dorsoventral muscles (DVM), which are responsible for the wing upstroke [1,66,67]. The DLMs form via templated-fusion and the DVMs form de novo, but both sets of muscles express Spalt major (Salm), the master regulator of the fibrillar muscle fate, and have the same fibrillar morphology in adult flies [6,73,75].

The musculature of *Drosophila* is a well-established model for understanding fiber-type specific differentiation [6,52,54,76]. Fibrillar and tubular adult fiber-types differ in both developmental origin and gene expression. For example, leg muscles develop from Lb (Ladybird) positive myoblasts of the leg disc [77]. Although both DFM and IFM derive from wing disc myoblasts, Cut-positive cells contribute to tubular DFM while vestigial (Vg)-positive cells contribute to fibrillar IFM [78,79]. Salm, which is induced downstream of Vg and Homothorax (Hth), is necessary and sufficient for the fibrillar fate, as loss results in tubular transformation of IFM and ectopic expression results in fibrillar transformation of leg muscle [73,80]. Salm directs a fibrillar-specific splicing program in IFM, in part through fiber-type specific induction of the RBP Bruno1 (Bru1) [75,81]. Alternative splicing is at least as important as transcriptional regulation in fiber-type identity, as based on mRNA-Seq data, 52% of structural genes are significantly differentially expressed and 55% are significantly differentially spliced between tubular and fibrillar fiber-types [6]. Fiber-type specific splice isoforms of structural genes such as *Myosin-heavy chain* (*Mhc*), *Projectin* (*bent, bt*), *Troponin-T* (*upheld, up*), *Myofilin* (*Mf*), *Tropomyosin 1* (*Tm1*) and *Strn-Mlck* (*Stretchin-Mlck*) confer the myofiber with distinct biophysical properties [6,62,63,82], demonstrating the physiological relevance of splice isoform expression. Only 23 RBPs have a characterized function in any type of *Drosophila* muscle [11], highlighting the need to systematically identify and study RBPs to understand muscle- and fiber-type specific regulation of splice isoform expression.

Here, we demonstrate the utility of the *Drosophila* model to investigate muscle-specific RBP function through a candidate RNAi screen. Our bioinformatic characterization of RBPs expressed in adult muscles identifies hundreds of diverse RBPs with temporal and spatially regulated expression dynamics, many of which additionally have muscle-specific RNAi phenotypes. We verify phenotypes for 35 of these RBPs in a candidate RNAi screen, uncovering a wide diversity in associated myofiber and myofibril defects in IFM. We find that phenotypes in many cases depend on RNAi knockdown efficiency, suggesting that muscle is sensitive to RBP expression levels. We also see phenotypic differences with temporally restricted RNAi knockdown. We uncover phenotypes for spliceosome components, indicating they may have additional regulatory functions beyond basal splicing activity during muscle development. We further characterize the knockdown phenotypes for *SF1* and *Hrb87F* in greater detail and show that both genes, which are known components of the spliceosomal A complex, regulate muscle-specific alternative splicing necessary for proper myofibril assembly, Z-disc integrity and kettin localization in IFM. Taken together, our data identify new RBPs with muscle-specific functions, provide insight into the regulatory function of individual spliceosome components and demonstrate the efficacy of *Drosophila* as a genetic model to study muscle-specific RNA regulatory dynamics.

## 2. Materials and Methods

A key resources table is provided in Appendix A, listing the sources of all genetic and chemical reagents and bioinformatic tools used in this manuscript.

### 2.1. Fly Stocks and Crosses

Fly stocks were grown on cornmeal medium and maintained under standard conditions. *w^1118^* was used as the wild-type control. RNAi lines used in the screen were obtained from the Vienna Drosophila Resource Center (VDRC), Bloomington Drosophila Stock Center (BDSC) or the National Institute of Genetics (NIG-FLY) and are listed in Appendix A. Gal4 drivers used include Mef2-Gal4 [83], which drives in all muscle; UAS-Dcr2, Mef2-Gal4 which enhances RNAi efficiency; and Act88F-Gal4 [84] which drives largely in indirect flight muscle (IFM) from 24h after puparium formation (APF). Mef2-Gal4 crosses were grown at 27 °C, while Act88F-Gal4 crosses were grown at 25 °C.

### 2.2. Selection of Candidate Genes for Screen

Genes for the candidate RNAi screen were selected based on several criteria. We selected 17 components distributed across the major spliceosome complexes as annotated in the Spliceosome Database [85] and with multiple available RNAi lines in public stock centers, A complex: *CG9346* (*U2SURP*), *Hrb87F* (*hnRNPA*), *SF1*, *Spf45* (*RBM17*), *snf* (*SNRPA*); B complex: *CG6686* (*SART1*), *CG6841* (*PRPF6*); A, B, C, P complex: *noi* (*SF3A3*), *Sf3a2*, *Sf3b1*, *Sf3b2*, *Sf3b3*, *Spx* (*SF3B4*); B, C, P complex: *Cdc5*, *fand* (*XAB2*, Prp19 complex), *Prp19* and *Prp8*. We also selected 18 RBP genes that are not direct spliceosome components. A genome-wide RNAi screen with Mef2-Gal4 identified knockdown of *CG9346* and *Rm62* to be flightless and knockdown of *Atx2*, *me31B*, *clu*, *Doa*, *SF1*, *bru1*, *how*, *mub*, *Prp8*, *sbr* and *Spx* to be lethal [86]. *bru1*, *Doa*, *mub*, *rump*, *Atx2* and *Rbfox1* are significantly differentially expressed among wild-type fibrillar IFM and tubular leg, TDT or *salm-IR* IFM [75,87]. Mbl, Me31B and Orb are identified as interactors of Bru1 in STRING [88,89]. *clu*, *how*, *Rm62*, *sbr* and *Smn* are significantly differentially expressed in *bru1-IR* IFM [75]. *Mettl14*, *Mettl3* (*Ime4*) and *Ythdc1* are identified components of the N-methyl-adenosine (m6A) modification pathway in flies [90]. *Rbp9* is a member of the Elav-like family of RBPs that regulates alternative splicing in neurons, and similar to Bru1, has three RRM domains [91].

### 2.3. Lethality and Behavior Assays

Flight behavior was tested as described previously (Schnorrer et al., 2010). Briefly, flies were introduced into the top of a 1-meter-long cylinder divided into 5 zones. Flies landing in the top two zones 4 and 5 were “normal fliers”, in the middle two zones 2 and 3 are “weak fliers” and at the bottom (zone 1) are “flightless”. The flight index used in Figure 2 and Figure 4 is defined as F_i_ = [(n_zone4_ + n_zone5_)/n_total_] – [(n_zone1_ + n_zone2_ + n_zone3_)/n_total_], where *n* = number of flies. We considered F_i_ < −0.75 as flightless, −0.75 < F_i_ < 0.75 as weak fliers and F_i_ > 0.75 as normal flight ability. Lethality stage was monitored and recorded as embryonic (no hatching larvae), early (no pupa formed), early pupal (lethal before P10), late pupal (pupae reach P13 or P14, but do not eclose) or pharate/adult lethal. Pupal eclosion (Figure 6) was determined by counting the number of flies that eclose from 80–100 pupae of the appropriate genotype.

### 2.4. Immunofluorescence and Microscopy

IFMs were dissected and stained as previously described [47]. After removing head and abdomen, thoraxes were fixed for 15–30 minutes in 4% PFA in 0.5% PBS-T (1× PBS + Triton-X100). Thoraces were then cut longitudinally with a microtome blade, blocked in 5% normal goat serum in PBS-T for 2 hours and stained overnight at 4 °C. Primary antibodies used included rabbit anti-GFP (1:500) and rat anti-kettin 1:100 (Babraham). Samples were washed three times in 0.5% PBS-T for 10 min and incubated overnight at 4 °C with secondary conjugated antibodies (1:500) from Invitrogen (Molecular Probes), including Alexa488 goat anti-rabbit IgG, Alexa647 goat anti-mouse IgG, and rhodamine-phalloidin. Samples were washed three times in 0.5% PBS-T and mounted in Vectashield containing DAPI. Confocal images were acquired on a Leica SP8X WLL and analyzed with Image J/Fiji [92]. Sarcomere length and width were measured in Fiji (Image J) based on rhodamine-phalloidin staining using the measure function because of the variability in myofibril structure after *SF1* and *Hrb87F* knockdown. Plotting and statistical analysis were performed in GraphPad Prism 9.

### 2.5. mRNA-Seq

IFMs were dissected from 72 h after puparium formation (APF) pupal thoraxes as described previously [93], producing samples highly enriched for IFM with low levels of contamination from tendon, motor neurons and trachea. Tissue was dissected for a maximum of 20–30 minutes and frozen in Trizol at −80 °C to minimize RNA degradation and changes to the transcriptome. IFMs from ~100 flies were combined into a single sample prior to total RNA isolation. Library preparation and sequencing was performed by LC Sciences (Houston, Texas, USA). Poly A-selected and stranded libraries were sequenced as Illumina paired-end 150 base pair reads with a depth > 60 million reads per library.

### 2.6. Bioinformatics

mRNA-Seq data from *SF1-IR* and *Hrb87F-IR* IFM was generated for this manuscript and is available from GEO under accession number GSE184001. All other mRNA-Seq data used in this manuscript have been published previously [57,75] and are available from GEO under accession numbers GSE63707, GSE107247 and GSE143430. For all datasets, reads were mapped (or remapped) with STAR to ENSEMBL genome assembly BDGP6.22 (annotation dmel_r6.32 (FB2020_01)), indexed with SAMtools and features counted with featureCounts. Analysis and visualization were performed in R using the packages listed in Appendix A. Soft clustering to identify temporal expression patterns was performed with Mfuzz [94]. Differential expression was analyzed on the gene level with DESeq2 [95] and on the level of exon use with DEXSeq [96]. GO term enrichments were performed with Gorilla [97] and term complexity reduced in R with rrvgo (https://ssayols.github.io/rrvgo, accessed on 15 July 2021).

For most analyses, the significance threshold included all exons with a *p* value < 0.01 or all genes with an adjusted *p* value < 0.01, to capture trends in the data independent of the magnitude of fold change. Plots with an additional fold change threshold are noted in the text and figure legends. Fiber-type specific genes were defined as all genes with a DESeq2 adjusted *p* value < 0.01 and abs(log_2_FC) > 1.5 in *salm-IR* IFM and leg or *salm-IR* IFM and jump muscle (TDT) as compared to wild-type (WT) IFM. Fiber-type specific exons were defined as all exons with a DEXSeq *p* value < 0.01 in *salm-IR* IFM and leg or *salm-IR* IFM and TDT as compared to WT IFM.

Gene sets of annotated RBPs were obtained from AmiGO [36,98], GLAD [35] or a paper that compiled RBPs identified by RNA-interactome capture (RIC) experiments [20]. A list of spliceosome components, including many RBPs, was obtained from the Spliceosome database [85]. Genome-wide phenotypic data for RBPs was obtained from a published RNAi screen [86]. A list of *Drosophila* sarcomeric proteins (SPs) has been compiled previously [57]. Exon classification as CDS, 5’-UTR or 3’-UTR was downloaded from Flybase.

## 3. Results

Muscle-specific functions are known for only a small portion of the hundreds of annotated RBPs in the eukaryotic genome [11]. To identify additional RBPs and splicing associated proteins in *Drosophila* that are expressed and thus potentially have a function in adult muscle, we looked for annotated RBPs in published mRNA-Seq datasets from fibrillar indirect flight muscle (IFM), tubular tergal depressor of the trochanter muscle (TDT, jump) and whole leg [57,75]. We considered four different lists of RBPs, including genes with the terms “RNA binding” or “mRNA binding” in AmiGO [36,98], genes identified as “RNA binding” in GLAD [35], a curated list of proteins identified by *Drosophila* RNA-interactome capture (RIC) experiments to bind RNA [20], and spliceosomal proteins listed in the Spliceosome database [85] (Appendix A, Appendix A). We found that 64% or more of RBPs contained in any one list are expressed in adult muscle (Appendix A). These curated lists vary in length and content and only partially overlap (Appendix A), reflecting the inherent challenge in identifying and classifying proteins as RBPs. We therefore conservatively define 568 “muscle RBPs” that are found in at least two annotations and are expressed in muscle (Figure 1A). Most muscle RBPs are expressed in all fiber types, as 95% (537 out of 568) can be found in all three (IFM, leg and TDT) mRNA-Seq datasets (Appendix A). We conclude that there are hundreds of RBPs expressed in muscle that may contribute to RNA regulation.

### 3.1. Muscle RBPs Are Expressed in Distinct Spatial and Temporal Patterns

IFMs are known to undergo dynamic shifts in gene expression that are correlated with the different phases in myofibril formation and sarcomere assembly, with a marked switch in expression as the myofibrils undergo maturation and establish stretch-activation properties [57]. To test if RBPs are regulated in a similar manner, we evaluated the temporal expression dynamics of muscle RBPs. RBPs display a dynamic range of expression level among adult muscle fiber types and throughout IFM development (Appendix A, E). Using a published dataset collected from eight timepoints across IFM development [57], we used Mfuzz [94] to define 9 RBP gene clusters with distinct temporal expression profiles (Figure 1B, Appendix A, Appendix A). These clusters are described by three main patterns: 1) RBPs with preferential expression in myoblasts or during early pupal development (16–48 h after puparium formation (APF)); 2) RBPs with peak expression during pre-myofibril formation and initial stages of myofibril assembly (30–48 h APF); 3) RBPs upregulated during later stages of myofibril maturation and in adult flies (48 h APF–1 d adult). These data illustrate that RBPs undergo temporal switches in expression and suggest that different sets of RBPs may be necessary for RNA processing at different stages of muscle development.

*Drosophila* have two recognized muscle fiber-types with distinct morphologies and functional properties, namely the fibrillar IFMs and the tubular body muscles [6,54]. Spalt major (Salm), the master transcriptional regulator of the fibrillar fate [73], induces expression of the RBP Bru1 in IFM to promote many fibrillar-specific alternative splice events [75,81]. To determine if other RBPs have fiber-type specific expression profiles, we evaluated RBP expression at the gene level using DESeq2 [95] (Figure 1C) and at the exon-usage level using DEXSeq [96] (Figure 1D, Appendix A). We found that 56% of muscle RBP genes (318 of 568) are significantly differentially expressed (DE) between fibrillar IFM and tubular TDT or leg muscle or have significantly altered expression levels in tubular-converted *salm^−/−^* IFM or *bru1* knockdown IFM (Figure 1C). 36% (206 of 568) of muscle RBP genes have at least one exon that is significantly differentially expressed among muscle fiber types or in *salm^−/−^* or *bru1-IR* IFM (Figure 1D). We observed that RBP expression levels and exon-usage patterns also differ between the TDT and leg samples, likely reflecting sub-specialization of tubular muscle types. A total of 138 genes shows both gene and exon-usage level regulation (Appendix A); thus, in total, 68% (386 of 568) of muscle RBPs are differentially expressed or differentially spliced among muscle fiber-types or upon loss of Salm or Bru1. Notably, 45 genes had exons significantly differentially regulated between IFM and tubular muscle in at least two comparisons (Figure 1E), displaying a clear pattern of muscle-type specific exon use, and thus likely muscle-type specific splice-isoform use. These RBPs are diverse and include splicing factors (i.e., *WDR79*, *qkr54B*, *Pep*, *Pcf11*, *mbl*), spliceosome components (i.e., *U2af38*, *Prp8*), nuclear export factors (*sbr*), ribosome components (i.e., *RpS4, RpS9, RpS20*, *RpL30*), and proteins associated with mRNA stability, localization or translation (i.e., *qkr54B*, *how*, *gw*, *eIF1A*, *egl*), suggesting that many RNA regulatory processes might be impacted by muscle-type specific gene or isoform expression. Considering that we could detect mRNA from most RBPs in all muscle types (Appendix A), these findings suggest that the regulation of RBP gene and splice–isoform expression levels plays an important role in the regulation of RNA processing in muscle.

### 3.2. RNAi-Mediated Knockdown of Hundreds of RBPs Produces Muscle Phenotypes

We next evaluated how many RBPs might have muscle-specific phenotypes. As a first look, we analyzed published data from a genome-wide RNAi screen using the muscle-specific Mef2-Gal4 driver [86]. A large portion of muscle RBPs (246 of 568, 43%) as well as of all annotated RBPs (665 of 1924, 35%) displayed a lethal phenotype, and another 5% (31 of 568, 98 of 1924) had impaired flight ability when knocked down specifically in muscle (Figure 1F, Appendix A, Appendix A). A smaller portion of RBPs with flight or lethality phenotypes had detectable myofiber defects (irregular or missing myofibers) or defects in myofibril or sarcomere structure (actin blobs, fuzzy Z-discs, degenerate or missing sarcomeres) (Figure 1F, Appendix A). IFMs were examined for RBPs with impaired flight or late pupal lethal phenotypes, while larval muscle was examined for lines with larval lethality phenotypes [86]. This suggests that 40–50% of RBPs, or somewhere between 280 to 773 different RBP genes, are required for proper muscle development. A large portion of these genes are also required for viability, either to enable larval hatching and locomotion or to facilitate adult eclosion.

### 3.3. A Candidate RNAi-Screen for RBP and Spliceosome Component Phenotypes in Muscle

In both vertebrates and invertebrates, alternative splicing (AS) serves as a mechanism to fine tune contractile properties through the production of fiber-type distinct isoforms of many sarcomere proteins [4,6,7], and thus plays an important role in both muscle development and maintenance. We were particularly interested in identifying RBPs that might be involved in fiber-type specific alternative splicing; thus, we selected a set of 35 candidate genes (Figure 2A, Appendix A, Appendix A) and performed a muscle-specific candidate RNAi screen using Mef2-Gal4. In addition to the hairpins used in the Schnorrer et al. screen [86]; we included the second-generation KK hairpins from VDRC as well as the TRiP hairpins available from Bloomington, and screened in total 105 RNAi lines (Appendix A). We assayed both lethality and flight ability, characterizing lethality phenotypes for 46 hairpins and impaired flight ability for an additional 40 hairpins (Appendix A). If we consider the phenotype of the strongest hairpin for each gene, 23 genes have lethal phenotypes and 9 genes have impaired flight ability (Figure 2B,C). A cross of the driver to *w^1118^* served as a wild-type control, and we used *salm-IR* [73] as a flightless control. We also observed a notable level of variation in phenotype among hairpins targeting the same gene (Figure 2C). Based on 76 hairpins, where we tested more than one hairpin per gene, we estimate that around 50% of hairpins produce efficient knockdown, leading to strong phenotypes, 30% of hairpins produce weak or moderate knockdown and weaker phenotypes, and around 20% of hairpins are ineffective (Appendix A). More generally, our candidate selection approach successfully enriched for RBPs with muscle function.

### 3.4. Muscle-Specific RBP Knockdown Leads to Diverse Myofibril and Sarcomere Defects

To evaluate myofiber and myofibril defects caused by RBP knockdown, we performed confocal microscopy with 35 hairpins targeting 17 RBP genes. We analyzed IFM morphology in 1 d old adults for crosses with impaired flight ability, and at 90 h after puparium formation (APF) for crosses leading to late pupal or pharate lethality. We observed a broad diversity in the severity and types of IFM phenotypes after RBP knockdown (Figure 3A). In severe cases, for example with knockdown of *Sf3a2* with line *55650* (*Sf3a2-55650*) (Figure 3C), IFM myofibers were completely missing. A missing myofiber phenotype at 90h APF can result from defective myoblast migration or fusion or impaired tendon attachment [48,49], as well as defective regulation of cellular growth and tension [55,58,60], which suggests that RBP function is necessary for one or more of these cellular processes. We frequently observed detached or torn myofibers, for example in *mbl-29585* (Figure 3D), *CG9346-27013 (U2SURP)* (Figure 3G), *bru1-41586* (Figure 4E), and *Rbfox1-27586* (Figure 3H). A less severe myofiber phenotype in several flightless lines was characterized by stretched and atrophic fibers, for example as observed with *Doa-50902* (Figure 3E) and *mub-34870* (Figure 3I). Both stretched and torn myofibers are commonly associated with hypercontraction, where misregulated myosin activity leads to aberrant contractility [99,100]. The Bru1 and Rbfox1 phenotypes served as positive controls, as we have previously characterized that loss of either protein leads to mis-splicing and hypercontraction phenotypes [75,87]. We conclude that on the fiber level, RBP knockdown can alter the number of myofibers and lead to myofiber hypercontraction, tearing or loss.

We next analyzed myofibril and sarcomere structure after RBP knockdown. Myofibrils in wild-type controls have a fibrillar structure (myofibrils are organized in parallel and not laterally aligned) with regular, crisply defined rectangular sarcomeres (Figure 3J,P). Myofibrils and sarcomeres after RBP knockdown displayed a variety of defects, from myofibril splitting and fraying to short sarcomeres and actin inclusions. Hairpins with the strongest myofiber defects also typically displayed severe myofibril and sarcomere defects. For example, IFM myofibers were frequently missing after knockdown with *Sf3a2-55650*, and remaining myofibrils, when not severely degraded, were split and disorganized with short sarcomeres (Figure 3K). Knockdown of *mbl*, *bru1* and *Rbfox1* (Figure 3O2,T,U), as we have previously reported [11,75,87], generated short sarcomeres and thick myofibrils, characteristic of hypercontraction. We saw similar defects for knockdown with *Rm62-46908* (Figure 3N1), *Rm62-110102* (Figure 3N2), *mub-34870* (Figure 3Q), *snf-330329* (Figure 3R), and *sbr-103715* (Figure 3S). This may indicate that these RBPs also regulate splicing of cytoskeletal components necessary for the proper regulation of myosin activity.

The most common defect we observed with Mef2-Gal4-mediated RBP knockdown, including in lines with largely normal flight ability, was abnormal actin inclusions that form at the Z-disc, or so-called “zebra bodies.” The frequency and size of zebra bodies increased as flight ability decreased. We observed these actin inclusions in lines where sarcomere structure appeared otherwise normal or merely thin, for example *Doa-46449* (Figure 3M), *clu-42318* (Figure 3X), *Prp6 (CG6841)-34254* (Figure 3Y), *Rbm17 (Spf45)-32949* (Figure 3Z) and *Hrb87F-52937* (Figure 3AA), as well as in combination with other myofibril and sarcomere defects as with *sbr-103715* (Figure 3S) or *Rbfox1-27586* (Figure 3U). In certain cases, it appears the formation of actin inclusions may potentially drive or at least precede myofibril splitting and fraying, for example *snf-330329* (Figure 3R) or *CG9346 (U2SURP)-27013* (Figure 3V). Taken together, we interpret these morphological data to reflect the diversity in RBP targets and RBP functions. Different RBPs regulate disparate aspects of muscle development, and therefore knockdown would be predicted to result in the observed phenotypic variability among RBP genes.

We also noted that phenotypic severity often differs among hairpins targeting the same gene. For example, although both *Rm62-46908* and *Rm62-110102* were pupal lethal, sarcomeres from *Rm62-46908* knockdown were short with sporadic, irregular actin densities (Figure 3N1), while sarcomeres from *Rm62-110102* knockdown were short and thick and myofibrils were frayed with actin blobs (Figure 3N2). A similar phenotypic variation was observed with *mbl* knockdown: *mbl-29585* was pupal lethal with zebra bodies (Figure 3O1); *mbl-28752* was pupal lethal with short, thick sarcomeres and partially degraded myofibrils (Figure 3O2); and *mbl-105486* was flightless with short, thick sarcomeres (Figure 3T). We propose this variability arises from target sensitivity to knockdown efficiency. Many RBPs are able to bind multiple or degenerate binding motifs with different levels of affinity [20]; thus, RBP function would be more or less impaired depending on knockdown efficiency.

### 3.5. Temporally Restricted Knockdown of RBPs in IFM Using Act88F-Gal4

Mef2-Gal4 is expressed throughout muscle development, including during embryonic and larval stages, which can lead to early lethality if RBPs are required in both embryonic and adult muscle. As nearly 50% of the hairpins in our screen were lethal when crossed to Mef2-Gal4, we performed a follow-up screen with Act88F-Gal4, which is reported to drive in IFM starting around 24h APF [84]. In theory, this should allow us to bypass lethality and assay muscle-specific phenotypes in IFMs, which are not required for adult survival. We screened 65 hairpins targeting 27 RBP genes with Act88F-Gal4 and assayed both lethality and flight ability (Figure 4A, Appendix A). We found that knockdown with 25 hairpins (38%) resulted in impaired flight ability, 13 (20%) produced no phenotype and 27 (42%) still resulted in lethality, although mostly late pupal or pharate lethality (Appendix A). Taking the strongest hairpin phenotype for each gene, knockdown of 8 RBPs (30%) impaired flight ability and knockdown of another 16 RBPs (59%) resulted in pupal lethality (Figure 4B), largely due to failed eclosion. This lethality phenotype occurs later than lethality observed with Mef2-Gal4, suggesting that knockdown is indeed restricted to the pupal stage. However, as IFMs are not required for eclosion, we checked if we could detect Act88F-Gal4 driver expression in tubular muscles which are required for eclosion. When crossed to a nuclear GFP, we could detect strong Gal4 expression in IFM as well as weak signal in a subset of fibers in adult jump, leg and gut muscle, but not in abdominal muscle (Appendix A–G). This implies that lethality phenotypes observed with Act88F-Gal4 likely reflect RBP requirement in these tubular muscles.

We next analyzed the morphology of IFM myofibers and myofibrils by confocal microscopy. We classified the phenotypes into different morphological categories, as defined above with Mef2-Gal4 knockdown (Figure 4C). Myofibers, as compared to the wild-type control (Figure 4D), were detached or torn, for example after knockdown with *bru1-41568* or *sbr-46117* (Figure 4E,G). Knockdown of *Atx2-108843* led to stretched myofibers (Figure 4F). We observed myofibril and sarcomere phenotypes including frayed or thin myofibrils with short, disorganized or at times degenerate sarcomeres. For example, knockdown of *bru1-41568*, *sbr-46117* and *how-37342* all resulted in thick myofibrils with short sarcomeres (Figure 4I–K), likely indicating hypercontraction. In addition, *sbr-46117* myofibrils were frayed (Figure 4J) and *bru1-41568* myofibrils were degenerate (Figure 4I). As compared to Mef2-Gal4, Act88F-Gal4 knockdown infrequently resulted in actin inclusions at the Z-disc, but more frequently resulted in thin myofibrils with otherwise normal sarcomeres, for example *Prp19-32865*, *snf-51459*, *fand-104186* and *Hrb87F-100732* (Figure 4L–O). We emphasize that Act88F-Gal4 expression, at least in IFM, can first be detected around 24h APF; thus, knockdown effects will only be evident later in pupal development after the target protein has turned over. Thus, these phenotypes indicate that RBPs are required for developmental processes related to mid- or late steps in myofibril and sarcomere growth, organization and maturation.

### 3.6. Diverse Phenotypes Associated with Altered Levels of Spliceosome Components

Although the loss of core spliceosome components is lethal, the efficiency of the splicing reaction as well as splice site recognition are sensitive to spliceosome component levels [101]. To ascertain how the expression level of spliceosome components might affect myogenesis, we compiled the data from 44 hairpins in our RNAi screen targeting 17 spliceosome components (Figure 2A,C, Appendix A, Appendix A), including confocal data from 27 hairpins. We tested components from different spliceosome complexes, including: *Hrb87F* and *SF1* (A complex); *snf* (U1); *Sf3a2*, *Sf3b1*, *Sf3b2*, *Sf3b3*, *Spx* and *noi* (U2); *Spf45 (Rbm17)* and (*U2SURP)* (U2 associated); *Cdc5*, *fand*, *Prp19* (Prp19 complex); *CG6686 (Sart1/SNU66), CG6841 (PRPF6)*, *Prp8* (tri-snRNP) (Appendix A). Components of the E, U1 and U2 complexes (A complex components) play important roles in splice site recognition and spliceosome assembly and positioning, while the U2, Prp19 and tri-snRNP complexes form the catalytic core after B-complex assembly and perform the splicing reaction (Appendix A).

As expected, based on the essential requirement for splicing in all cells, we observed that muscle-specific knockdown of many spliceosome components is lethal. A total of 50% (22 of 44) of hairpins crossed to Mef2-Gal4 were lethal in early pupal or pre-pupal stages and 62% (18 of 29) of hairpins crossed to Act88F-Gal4 were lethal at late pupal stages (Figure 5A, Appendix A). We observed that only 15% of hairpins (2 of 13) targeting A-complex components as compared to 65% of hairpins (20 of 31) targeting B-complex components had lethal phenotypes when crossed to Mef2-Gal4 (Appendix A). We observed the opposite with impaired flight ability, where 38% of hairpins (5 of 13) targeting A-complex components as compared to 6% of hairpins (2 of 31) targeting B-complex components were flight impaired (Appendix A). One possible explanation for this result that remains to be tested in future experiments is that loss of specific E or A-complex components may lead to alternative splicing defects that impair muscle development and function, while loss of B-complex components results in a general block of splicing.

We also observed diverse myofibril and sarcomere defects after knockdown of spliceosome components. Phenotypes ranged from split and frayed myofibrils with degenerate sarcomeres (Figure 5D,K), to actin blobs at the Z-disc (Figure 5E,F,H,I), to thin myofibrils (Figure 5G,J,L–P). Notably, all myofibrils from Act88F-Gal4 knockdown of spliceosome components were thin (Figure 5A). By contrast, Act88F-Gal4 mediated knockdown of non-spliceosome associated RBPs such as *bru1-41568, sbr-46117* or *how-37342* (Figure 4I–K) resulted in thick myofibrils with short sarcomeres, in addition to myofibril tearing and abnormal actin structures. This suggests that decreased levels of core spliceosome components during IFM maturation may generally reduce mRNA levels and thus translation of most sarcomere components, while disrupting splicing factors alters isoform levels of specific sarcomere proteins and biomechanical properties of sarcomeres.

For four spliceosome components including *SF1*, *Hrb87F*, *snf* and *Prp19*, we were able to evaluate myofibril phenotypes after knockdown with three or more independent hairpins (Figure 5E–P). We observed that in several cases, different hairpins targeting the same gene give different apparent phenotypes. For example, Mef2-Gal4 mediated knockdown of *SF1-13425* and *SF1-13426* results in actin accumulations at the Z-disc, torn myofibrils and aberrant actin distribution along the myofibril, while Act88F-Gal4 mediated knockdown of *SF1-52938* results in thin myofibrils (Figure 5E–G). Similar defects are observed with both *Hrb87F* (Figure 5H–J) and *snf* (Figure 5K–M). We propose two possible explanations. First, the level of knockdown efficiency might vary among the hairpins, resulting in phenotypes of different severity. Second, knockdown with Mef2-Gal4 at early stages of muscle development may impair pre-myofibril formation leading to myofibril splitting, fraying, tearing and abnormal actin accumulations, while knockdown with Act88F-Gal4 rather impairs myofibril growth. Future studies will be required to clarify the detailed developmental mechanism.

### 3.7. SF1 and Hrb87F Are Necessary for Myofibril Development and Z-disc Integrity

We selected *SF1* and *Hrb87F* for further characterization, as neither gene has been previously characterized in *Drosophila* muscle and both genes are reported to be A-complex components that contribute to splice site recognition (Appendix A). More specifically, SF1 is responsible for branch-point sequence recognition [102,103], while Hrb87F, as a homolog of the hnRNPA1 family, is involved in 3’-SS recognition [104,105]. For each gene, we selected the hairpin with the strongest flight phenotype that was still viable when crossed with Mef2-Gal4, *SF1-13426* and *Hrb87F-31244* (Figure 2C, Appendix A, Appendix A), and examined the IFM phenotype in 1 d old adults. All six IFM myofibers were present and attached after knockdown of *SF1-13426* and *Hrb87F-31244*, although *Hrb87F-31244* myofibers had an uneven actin density and appeared stretched (Figure 6A,C,E). At a higher magnification, myofibrils from both *SF1-13426* and *Hrb87F-31244* were thick with aberrant sarcomere structure. In *SF1-13426* flies, the myofibrils were torn and frayed, and Z-discs as marked with kettin staining were wavy or bulged (Figure 6D1–D4). In *Hrb87F-31244* flies, the myofibrils were thick and frequently split with prominent zebra-bodies often spanning 2-3 myofibrils (Figure 6F1–F2). Kettin staining revealed that Z-discs were spotty and in the vicinity of zebra-bodies discontinuous (Figure 6F3–F4). IFM sarcomeres in *Hrb87F-31244* flies are significantly shorter than the wild-type control (Figure 6G), while myofibrils from both *SF1-13426* and *Hrb87F-31244* are significantly thicker than the wild-type control (Figure 6H). These phenotypes are similar but not identical, suggesting that both SF1 and Hrb87F mediate splicing necessary for proper myofibril development.

One or more of the hairpins we tested against *SF1* and *Hrb87F* resulted in lethality (Figure 2C), indicating that *SF1-13426* and *Hrb87F-31244* mediate only a partial knockdown of the target gene. UAS-Dcr2, Mef2-Gal4 was previously reported to enhance RNAi knockdown efficiency [106]; thus, we next tested how a stronger knockdown with the same hairpins influenced the muscle phenotypes. While >90% of *SF1-13426* and *Hrb87F-31244* Mef2-Gal4 knockdown pupae eclose as adults, both hairpins are pupal lethal when crossed to UAS-Dcr2, Mef2-Gal4 (Figure 6I). This confirms that Dcr2 enhances knockdown efficiency and phenotypic severity.

We then examined IFM phenotypes at 72 h APF, a timepoint where sarcomere structure is well defined but before knockdown induced pupal death. We found that IFM myofiber structure is severely disrupted, with the majority of fibers torn or missing with both *SF1-13426* and *Hrb87F-31244* (Figure 6J,M,P). Myofibril and sarcomere structure in remaining fibers are severely affected. Myofibrils in IFM from *SF1-13426* flies are thin and frequently split or frayed (Figure 6N1–N3). Sarcomeres have notable Z-disc defects, including misorientation, abnormal actin accumulation and poorly refined structure (Figure 6N2). Kettin localization was severely impaired with the majority of kettin staining localizing to cytoplasmic puncta (Figure 6N3). Myofibrils in *Hrb87F-31244* IFM were split, frayed and torn with poorly defined sarcomeric structures (Figure 6Q1–Q3). Kettin was mostly localized to cytoplasmic puncta (Figure 6Q3). Myofibril width as well as sarcomere length was significantly shorter with *SF1-13426* and *Hrb87F-31244* knockdowns at both 72 h APF and 88 h APF (Figure 6G,H). As these flies are unable to eclose, we also examined tubular leg muscle structure. Myofibrils in tubular leg muscle are disorganized with poorly defined sarcomeres in *SF1-13426* and *Hrb87F-31244* flies (Figure 6L,O,R). We conclude that SF1 and Hrb87F are required for myofibril development and likely for Z-disc integrity, and a stronger knockdown enhances the severity of myofibril and Z-disc phenotypes. We also infer that fibrillar IFM is more sensitive to alterations in expression levels of both proteins, as functional and morphological defects in tubular leg muscle are first observed with Dcr2-enhanced knockdown.

### 3.8. Transcriptomic Analysis Reveals Molecular Defects in SF1 and Hrb87F Knockdown IFM

To characterize the transcriptomic phenotype and identify possible developmental mechanisms that disrupt myofibril structure after knockdown with *SF1-13426* and *Hrb87F-31244*, we performed mRNA-Seq. We dissected IFM at 72h APF from Mef2-Gal4 driven knockdown flies, as well as from Mef2-Gal4 crossed to *w^1118^* as a wild-type (WT) control. We analyzed differential gene expression between knockdown and WT IFM using DESeq2 [95] and differential exon use with DEXSeq [96]. Differential exon use reflects differential mRNA isoform use due to either alternative splicing or alternative promoter use. We also specifically examined a set of previously annotated genes encoding *Drosophila* sarcomere proteins [57] in these data. We were able to detect changes in both gene expression (Figure 7A,B) and exon use (Figure 7D,E) in IFM after knockdown with *SF1-13426* and *Hrb87F-31244*.

We started by characterizing transcriptomic defects in *SF1-13426* (*SF1-IR*) and *Hrb87F-31244* (*Hrb87F-IR*) IFM at the gene level. Although knockdown of *SF1* leads to both up- and downregulation of differentially expressed (DE) genes, sarcomere genes tend towards weak downregulation (Figure 7A, Appendix A). Sarcomere genes as well as DE genes in general tend to be weakly downregulated in *Hrb87F* knockdown IFM (Figure 7B). The small fold changes we observe may be the result of a hypermorphic phenotype due to partial RNAi knockdown or may alternatively reflect an overall decrease in all mature mRNA. We performed a gene ontology (GO) analysis of enriched terms in up- or downregulated genes. Upregulated genes in *SF1-IR* are enriched in biological process terms such as “response to stress,” “mature ribosome assembly” and “telomere organization,” while downregulated genes are enriched for terms including “adult somatic muscle development,” “myofibril assembly,” “translation,” “endoplasmic reticulum unfolded protein response,” and “regulation of mitochondrial organization” (Figure 7C). Upregulated genes in *Hrb87F-IR* are enriched for “developmental process” and “cytoskeleton organization” terms, while downregulated genes are enriched for “myofibril assembly,” “cytoplasmic translation,” “ATP metabolic process” and “mitochondrion organization,” among others (Figure 7C). This suggests that both SF1 and Hrb87F regulate muscle structural genes among many other targets, and a decrease in SF1 and Hrb87F levels more generally leads to a decrease in translation and mitochondrial function and activation of the stress response, which remains to be verified in future experiments.

We then characterized changes in exon use in *SF1-13426* (*SF1-IR*) and *Hrb87F-31244* (*Hrb87F-IR*) IFM. Knockdown of both *SF1* and *Hrb87F* results in expression changes in hundreds of exons (Figure 7D,E, Appendix A). Notably, we observed up- and downregulation of different sets of sarcomere gene exons (Figure 7D,E, Figure 8A) that are comparatively larger than overall changes in sarcomere gene expression (Appendix A). This is further reflected in the enrichment for GO Biological Process terms such as “muscle system process,” “myofibril assembly” and “flight” in upregulated *SF1-IR* exons and “cytoskeleton organization” and “muscle structure development” in *Hrb87F-IR* downregulated exons (Figure 7C). We observed exons from the same sarcomere gene regulated in opposite directions. For example, one set of exons from *bt*, *sls*, *Mhc*, *Ryr*, *tn*, *Mf* and *Zasp66* are upregulated, while a different set of exons is concurrently downregulated (Figure 8A), indicating that ratios of splice isoform expression in IFM for multiple sarcomere components are altered when levels of SF1 and Hrb87F are reduced. As fiber-types have distinct isoform expression profiles [6], splicing alterations likely contribute to the observed defects in IFM structure and function.

As our immunostaining of both *SF1-IR* and *Hrb87F-IR* IFM revealed mislocalization of kettin (Figure 6D3,F3,N3,Q3), we evaluated expression changes in Z-disc components that directly bind to or influence kettin localization including *bt* (Projectin), *sls*, *Zasp52*, *Zasp66*, *Zasp67*, *cher* (*filamen*) and *zormin* [107,108,109] (Appendix A). The *sls* gene, which encodes the kettin protein, is expressed at similar levels in *SF1-IR* and WT IFM (FC = 0.17, adjusted *p* value 0.6627) and at a slightly higher level in *Hrb87F-IR* IFM (FC = 0.63, adjusted *p* value 0.0017). Although *Zasp66* expression is weakly but significantly decreased in both *SF1-IR* and *Hrb87F-IR* and *Zasp67* is slightly decreased in *SF1-IR*, changes in gene-level expression of *bt*, *Zasp52*, *zormin*, and *cher* are non-significant (Appendix A, Appendix A). While we do not detect strong changes in gene expression, we do see changes in exon use within the kettin encoding exons of *sls* as well as in *bt*, *Zasp52* and *Zasp66* in both *SF1-IR* and *Hrb87F-IR* (Appendix A). These differences would likely alter the ratio of Z-disc protein isoforms in the IFM sarcomere, but it remains to be tested if any exon-use changes detected here are sufficient for kettin mislocalization, or if a different mechanism is responsible. Taken together, our mRNA-Seq analysis identified altered gene expression and exon use in *SF1-IR* and *Hrb87F-IR* IFM that is predicted to affect muscle structure, sarcomere contractility, mitochondrial function and metabolism.

### 3.9. SF1, Hrb87F and Bru1 Regulate Distinct but Overlapping Splice Events

SF1 and Hrb87F are both A-complex components required for splice site recognition, but one recognizes the branch point and the other the 3’-SS. To test if their splicing signatures are distinct as well as how they differ from a non-spliceosome associated muscle-specific splicing factor, we compared gene expression and exon use patterns in *SF1-IR*, *Hrb87F-IR* and *bru1-IR*. Bruno1 (Bru1) is a CELF-family homologue known to regulate alternative splicing important for sarcomere assembly and contractility in IFM [75,81]. We started by calculating the Pearson coefficient for pairwise comparison of the significant fold change values and found a weak positive correlation of 0.3666 between *SF1-IR* and *Hrb87F-IR* at the gene level, which increased to 0.6956 at the exon level (Figure 8B). By comparison, *bru1-IR* has little or no correlation to *SF1-IR* or *Hrb87F-IR*. Genes that are downregulated in *SF1-IR* are also downregulated in *Hrb87F-IR*, but genes that are upregulated in *SF1-IR* are largely unchanged in *Hrb87F-IR* (Appendix A). As Bru1 contributes to the regulation of fiber-type identity, we also looked at genes that are significantly differentially expressed between tubular and fibrillar muscle. Although *bru1-IR* leads to upregulation of tubular genes and downregulation of fibrillar genes, *SF1-IR* and *Hrb87F-IR* do not (Appendix A), indicating they are not involved in regulation of fiber-type identity.

We next evaluated exon usage patterns in the DEXSeq data. The majority of exons that are significantly changed in *SF1-IR* and *Hrb87F-IR* are regulated in the same direction (Figure 8C, Appendix A). This can also be clearly observed for sarcomeric gene exons, where *SF1-IR* and *Hrb87F-IR* have a similar regulatory pattern which is distinct from that of *bru1-IR* (Figure 8A). We see regulation of both coding exons as well as 5’- and 3’-UTR exons, indicating these changes reflect bona fide splicing changes as well as use of alternative promoters (Figure 8A). If we look at all significantly DE exons, we find only five events regulated by all three proteins, and a majority of events selectively regulated by one of the proteins (Figure 8C). Events regulated by Bru1 are largely distinct from those sensitive to decreased expression of SF1 and Hrb87F. For the small number of events significantly regulated in both *bru1-IR* and either *SF1-IR* or *Hrb87F-IR*, we observe much larger fold-changes in the *bru1-IR* IFM (Figure 8C). This is also observed when we look selectively at exons differentially regulated between tubular and fibrillar muscle (Appendix A), indicating that SF1 and Hrb87F do not play a large role in fiber-type specific splicing. Exons regulated in both *SF1-IR* and *bru1-IR* change in the same direction, while exons regulated in both *Hrb87F-IR* and *bru1-IR* tend to change in opposite directions (Figure 8C, Appendix A). These patterns indicate that RBPs that regulate splicing at the same step will have more similar splicing profiles than those that regulate splicing at a different step in the splicing process (Appendix A). However, individual exons and genes can be more or less sensitive to decreased expression of a specific RBP. A large number of fiber-type specific exons (148 of 216 in this dataset) are not significantly altered in *bru1-IR*, *SF1-IR*, or *Hrb87F-IR* (Appendix A). This may reflect use of a stringent *p* value threshold or developmental differences in fiber-type specific splicing between 72 h APF and 1 d adult, which we used to identify fiber-type specific exons. Moreover, this likely indicates that additional, uncharacterized RBPs contribute to fiber-type specific splicing and motivates future research into RBP function in muscle.

## 4. Discussion

RNA binding proteins (RBPs) play important roles in tissue development through their regulation of mRNA alternative splicing, trafficking, localization and stability. Their coordinated regulation fine-tunes ratios of splice-isoform expression and helps define muscle-type specific morphology and contractility [6,63]. Here, we provide insight into the breadth of this RNA regulatory network by identifying 568 RBPs expressed in transcriptomic data from *Drosophila* adult muscle and revealing their patterns of temporal and fiber-type specific expression. We screened 105 RNAi hairpins targeting 35 RBPs for muscle-specific phenotypes, including their effects on viability, flight ability, and the morphology of IFM myofibers, myofibrils and sarcomeres. Our data illustrate the diversity of muscle phenotypes that can be attributed to loss or altered levels of RBP expression and identify previously uncharacterized flight muscle-related functions for clueless (Clu), darkener of apricot (Doa), heterogenous nuclear ribonucleoprotein at 87F (Hrb87F), maternal expression at 31B (Me31B), mushroom-body expressed (Mub), Rm62, small bristles (Sbr), splicing factor 1 (SF1), and splicing factor 3a subunit 2 (Sf3a2). We examined developmental phenotypes for SF1 and Hrb87F in greater detail, showing that exon use profiles of sarcomeric genes are misregulated upon knockdown of both *SF1* and *Hrb87F*. This results in severe developmental defects in myofibril and sarcomere structure, including impaired localization of kettin to the Z-disc. Gene expression and exon use profiles in *SF1-IR* and *Hrb87F-IR* are largely distinct from those in *bru1-IR*, illustrating that splicing regulators have select targets and control discrete portions of the alternative splicing network in muscle.

### 4.1. Muscle Fiber-Type Specific Expression and Function of RBPs

Differential splicing of structural proteins among muscle fiber-types in both flies and vertebrates has been previously described, and is thought to underly their distinctive morphologies and contractile properties [6,7,82]. One mechanism to achieve fiber-type specific splicing patterns is through expression of fiber-type specific splicing factors. An example from *Drosophila* is Bruno1 (Bru1), a CELF-family homolog, which is predominately expressed in IFM and regulates fibrillar-specific splicing [75,81]. However, our data suggest that the majority of RBPs are expressed in all types of muscle (Appendix A). This is similar to RBP expression in vertebrates, where most splicing factors are expressed in all skeletal muscle fiber-types and achieve fiber-type specific function through a combination of isoform-specific expression, regulated subcellular localization or post-transcriptional modification [4,110]. For example, MBNL1 isoforms containing exon 7 are preferentially localized to the nucleus and inclusion of exon 5 further enhances splicing regulatory activity [111]. Such regulation is also disease relevant, as for example CUG repeat expansions found in patients with myotonic dystrophy sequester MBNL1 resulting in the abnormal stabilization of CELF1, disrupting normal RBP function [112,113,114]. We also observe differential use in RBP exons among muscle fiber-types in the fly (Figure 1D,E), suggesting basic mechanisms regulating RBP localization and function may be conserved and experiments in the fly model have the potential to expand our understanding of fiber-type specific RBP regulation.

### 4.2. Developmental Regulation of RBP Expression and Function during Myogenesis

Our data further indicate that there are developmental shifts in RBP expression and function during IFM myogenesis. This is evidenced by the clustering of distinct RBP temporal expression profiles (Figure 1B, Appendix A), as well as by differences in Mef2-Gal4 and Act88F-Gal4 driven knockdown phenotypes (Figure 2, Figure 3, Figure 4 and Figure 5). Temporal shifts in RBP expression levels and activity have been characterized in vertebrate muscle, notably with CELF-family proteins promoting embryonic splicing events and MBNL-family members promoting mature, post-natal splicing patterns [31,33]. Alternative splicing moreover refines contractile properties as well as transcriptional programs during myogenesis [115]. For example, a developmentally regulated switch in alternative splicing of the inhibitory MEF2Dα1 and MEF2Cα1 isoforms to the MEF2Dα2 and MEF2Cα2 isoforms that activate muscle-gene expression is necessary during vertebrate skeletal muscle differentiation, and this switch is disrupted in muscle diseases including myotonic dystrophy [116,117,118]. A transcriptional switch in *Drosophila* mediated by Spalt major (Salm) [73] as well as E2F [119] has been characterized to promote maturation of IFM, including regulation of the splicing network that favors fibrillar-specific events [75]. Additionally, alternative splicing of “early” and “mature” forms of sarcomere proteins such as Myosin heavy chain (Mhc) [120] and Zasp52 [121], among others, are reported in IFM, suggesting that transcriptional and alternative splicing switches are likely a conserved feature of muscle differentiation. It will be interesting to determine through future experiments how temporally regulated RBPs identified in our analysis contribute to the differentiation process.

### 4.3. Diversity in Muscle Phenotypes Likely Reflects RBP Target Specificity and Expression Level

A salient observation from our screen was the wide diversity in RBP-associated myofibril defects and the phenotypic dependence on knockdown strength. We showed that RNAi knockdown of RBPs can result in actin inclusions, torn, frayed, split, thick or thin myofibrils, short sarcomeres and disorganized or discontinuous Z-discs (Figure 3, Figure 4 and Figure 5). This phenotypic diversity most likely reflects the functional diversity of RBPs as well as their target specificity. As we show by comparing transcriptomic signatures of *bru1-IR*, *SF1-IR* and *Hrb87F-IR* (Figure 8 and Appendix A), different RBPs regulate distinct, but often partially overlapping, subsets of exons. This has been systematically demonstrated by a screen of 56 *Drosophila* RNA binding proteins in the Schneider 2 (S2) cell culture line derived from late-stage embryos that found extensive combinatorial regulation of target exons and notably wide-spread cross-regulatory interactions among RBPs [24]. Extensive combinatorial and cross-regulatory interactions, in addition to factor-specific targets, are also observed in vertebrate muscle, for example among CELF, MBNL and RBFOX family proteins [25,32]. This implies that muscle RBPs participate in carefully balanced regulatory networks and provides a possible mechanism for the RBP level-dependent regulation we observe: target exons could be bound by different RBPs with different affinities, and competitive or cooperative interactions with other RBPs are likely to depend on expression levels. RBPs that function at overlapping nodes in this network would be predicted to have similar phenotypes and potentially cross-regulate, as we have recently shown in a preprint for Bru1 and Rbfox1 [87]. Here we find that similar to Bru1 and Rbfox1, knockdown of *muscleblind* (*mbl*) and *held out wings* (*how*), previously identified to play a role in muscle development [11,122,123], as well as knockdown of *Rm62*, *mushroom-body expressed* (*mub*), *sans fille* (*snf*) and *small bristles (sbr*), produce similar hypercontraction-like phenotypes characterized by short sarcomeres and thick myofibrils (Figure 3 and Figure 4). Further experiments, including cross-linking immunoprecipitation (CLIP) to identify RBP direct targets and binding-site preferences, will be necessary to define the alternative splicing network in IFM and determine if and how these additional RBPs cooperate with Bru1 and Rbfox1 to regulate structural gene expression in muscle.

### 4.4. Knockdown of Spliceosome Components Results in Alternative Splicing Defects in Muscle

In addition to splicing regulatory factors, we evaluated 17 spliceosome components in our screen. Our data show that muscle specific knockdown with Mef2-Gal4 led to early lethality as well as severe defects in muscle structure, in particular for B- and C-complex members (Figure 5, Appendix A). By contrast, knockdown restricted to later stages of muscle development using Act88F-Gal4 resulted predominately in thin myofibrils. Knockdown of A complex components caused less lethality but disrupted both myofiber and myofibril structures. Our results suggest both temporal and knockdown-level dependent phenotypes of spliceosome components in IFM. There is precedence for this observation, as several core spliceosome components have been reported to regulate alternative splicing in addition to serving as components of the canonical spliceosome, and spliceosome components including SF3B1, U2AF1, SRSF2 and hnRNPA1, among others, are often targeted by somatic mutation in cancer [124,125,126]. Altered levels of snRNAs themselves produce defects in alternative splicing [127]. Levels of snRNPs and spliceosome complexes are decreased in spinal muscular atrophy (SMA), resulting in alternative splicing defects and contributing to disease pathology in both muscle and neurons [38,126]. Together with our data, these observations blur the line between “basal” spliceosome components and “regulatory” splicing factors, implying a far more dynamic regulation of splicing in muscle.

Knockdown of A-complex members *SF1* and *Hrb87F* results in myofibril defects including splitting, fraying, abnormal Z-disc structures and impaired localization of kettin, and these phenotypes can be enhanced using stronger knockdown conditions (Figure 6). Related sarcomere phenotypes have been previously characterized for proteins whose loss specifically compromises Z-disc integrity, for example Zasp52 [108,121,128], Zasp66 [107], filamin [129] or zormin [109,130]. We find that knockdown of both *SF1* and *Hrb87F* resulted in specific defects in sarcomere gene expression and exon use, including in *Zasp66* and *Zasp52* (Figure 7 and Figure 8). However, the regulated exons were largely distinct from events altered in *bru1-IR* and did not contribute significantly to fiber-type specific exon use or gene expression (Figure 7 and Figure 8). This shows that both factors mediate alternative splicing and have broad effects on muscle-specific gene expression, but SF1 and Hrb87F function in parallel to Bru1 and other fiber-identity factors in the splicing network.

The defects we observe in *Hrb87F-IR*, a *Drosophila* homolog of the hnRNPA family, are reminiscent of phenotypes in *hnRNPA1* knockout mice. These mice are embryonic lethal due to defects in heart and skeletal muscle, most prominently aberrant splicing of both muscle structural genes and key regulators of myofiber differentiation including *mef2c*, *notch4, Myo5a, Myo6, Pax6,* and *Foxo3*, among others [131]. Conditional knockout of *hnRNPA1* in mouse muscle promotes insulin resistance and more generally disrupts glucose metabolism [132], a parallel to GO enrichments for metabolic terms we see in DE genes in *Hrb87F-IR*. hnRNPA1 is also abnormally expressed in myotonic dystrophy, where similar to CELF1, it promotes fetal splicing patterns and antagonizes MBNL activity [133]. Our data on *Drosophila* Hrb87F are thus consistent with a conserved role for hnRNPA family members in regulating splicing during muscle development. Although SF1 has been characterized to bind the branch point site and regulate alternative 3’-SS selection [134,135], to the best of our knowledge, this is the first report of a striated muscle-specific splicing phenotype for SF1. In summary, our results illustrate the power and versatility of *Drosophila* as a well-conserved genetic model to identify factors with RNA-regulatory functions in muscle and to explore basic, disease relevant mechanisms of RBP function in myogenesis.

## Figures and Tables

**Figure 1 cells-10-02505-f001:**
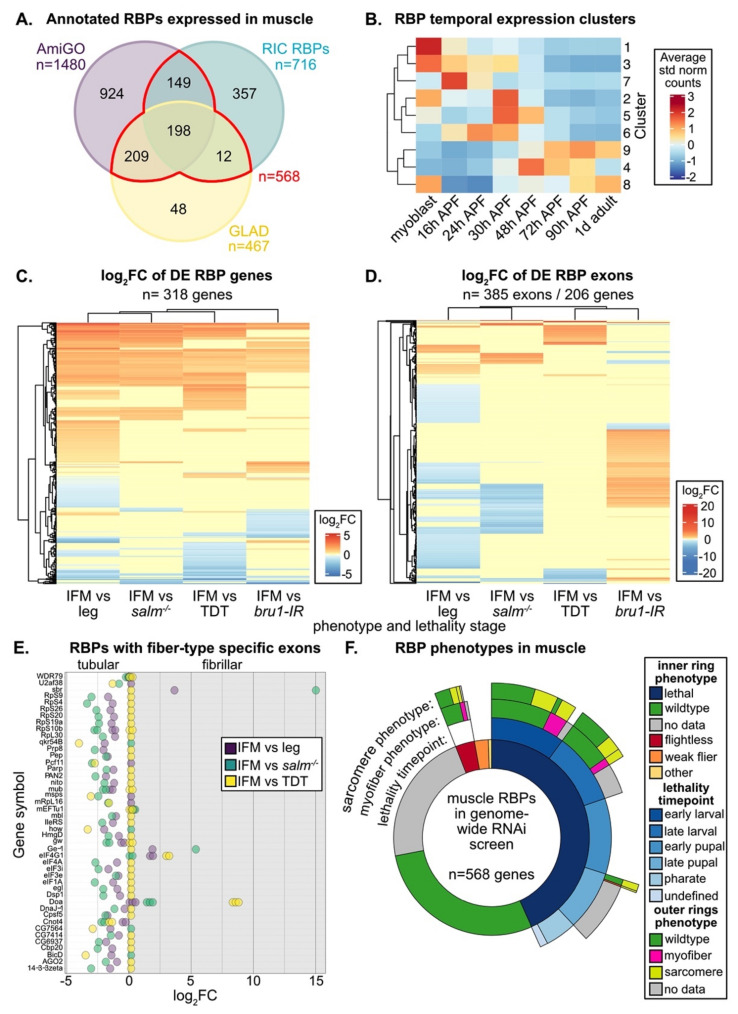
RNA binding proteins (RBPs) have fiber-type and temporal dependent expression patterns in muscle. (**A**) Comparison of RBPs from AmiGO (purple), GLAD (yellow) or RNA-interactome capture (RIC, cyan) annotations that are expressed in mRNA-Seq data from 1 d adult *Drosophila* indirect flight muscle (IFM), tergal depressor of the trochanter (TDT) or leg. A core set of 568 RBPs (red) are expressed in muscle and present in at least two annotations. (**B**) Heatmap of RBP temporal expression profiles identified from Mfuzz clustering of IFM mRNA-Seq timecourse data. Core expression profiles, the average of standard normalized count values for each cluster, cluster into three distinct temporal expression patterns. (C) Heatmap and hierarchical clustering of log_2_ fold change (log_2_FC) values from DESeq2 differential expression analysis of RBP genes between wild-type IFM and leg, TDT, *salm^−/−^* IFM or *bru1-IR* IFM. (**D**) Heatmap and hierarchical clustering of log_2_FC values from DEXSeq differential exon usage analysis of RBP gene exons between wild-type IFM and leg, TDT, *salm^−/−^* IFM or *bru1-IR* IFM. (**E**) RBP genes with exons that are differentially expressed between tubular and fibrillar fiber types. Exons with positive log_2_FC values are fibrillar preferential, while negative log_2_FC values are tubular preferential, as detected in DEXSeq comparison of wild-type IFM and leg (purple), TDT (yellow) or *salm^−/−^* IFM (cyan). (**F**) Sunburst plot of RBP phenotypes in muscle from a published genome-wide RNAi screen [86]. The inner ring summarizes lethality and flight phenotypes. The second ring depicts the stage of lethality. The third and fourth rings summarize myofiber (magenta) and sarcomere (yellow) phenotypes, respectively.

**Figure 2 cells-10-02505-f002:**
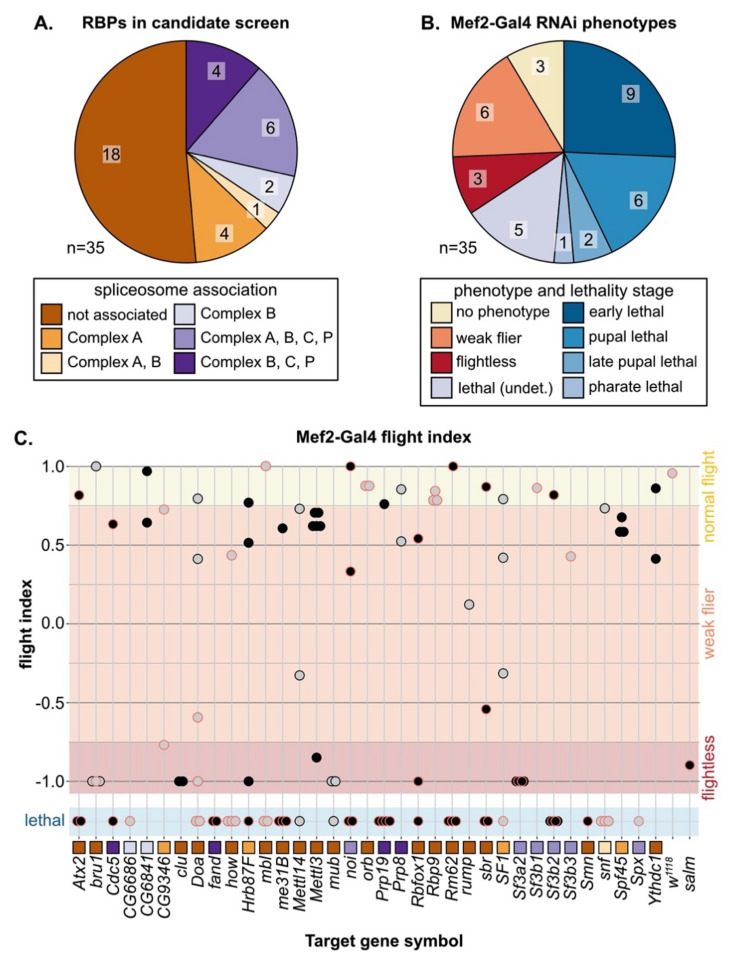
Muscle-specific RNAi knockdown candidate screen of 35 RBP genes. (**A**) Thirty-five RBPs were selected as candidates for further analysis in muscle, including 17 spliceosome associated genes. (**B**) Mef2-Gal4 knockdown lethality (blue shades) and flight (flightless, red; weak flier, orange) phenotypes for RBP genes based on the phenotype of the strongest RNAi hairpin. (**C)** Plot summarizing Mef2-Gal4 phenotype for 105 hairpins targeting 35 RBP genes. Dots represent individual hairpins, and a red outline represents hairpins additionally tested with Act88F-Gal4 (Figure 4). Flight ability was classified as normal (yellow), weak/impaired (orange) or flightless (red). Lethal hairpins (blue) were not tested for flight ability. Hairpins are organized into columns by gene. Squares indicate spliceosome membership and are colored as in (**A**).

**Figure 3 cells-10-02505-f003:**
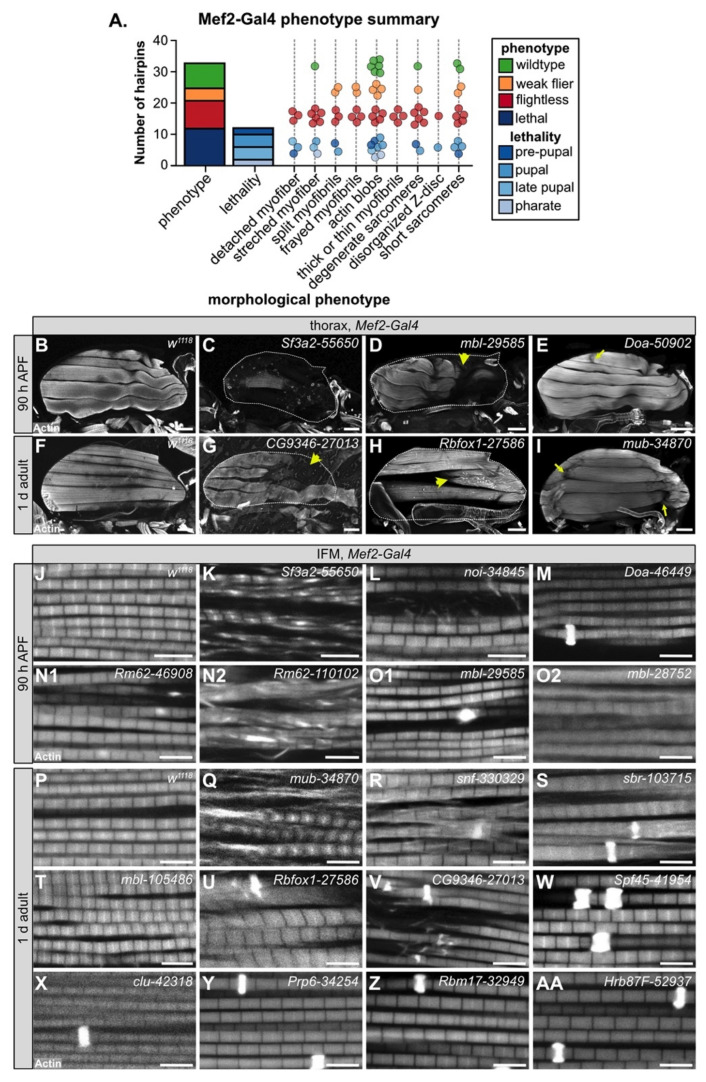
Mef2-Gal4 knockdown of RBPs produces diverse myofibril and sarcomere phenotypes. (**A**) Graphical summary of confocal phenotypes from 35 hairpins targeting 17 RBP genes. Dots represent hairpins and are colored by flight (flightless, red; weak flier, orange) and lethality (blue shades) phenotype. Hairpins may appear in more than one morphological phenotype category. (**B**–**I**) Confocal Z-projections of IFM hemi-thoraxes showing myofiber phenotypes at 90 h APF (**B**–**E**) or in 1 d adult (**F**–**I**). Torn, detached or missing fibers are observed in severe examples (yellow arrow heads in **D**,**E**,**G**,**H**) and classified as “detached myofiber” in (**A**). Other myofibers are stretched (yellow arrows in (**E**,**I)**), likely reflecting hypercontraction. White dotted lines mark boundary of the thorax in (**C**,**D**,**G**) and (**H**). Scale bars = 100 μm. (**J**–**AA**) Confocal single-plane images showing myofibril and sarcomere structure at 90 h APF (**J**–**O2**) or in 1 d adult (**P**–**AA**). As compared to *w^1118^* control images (**J**,**P**), knockdown of RBPs can result in split or frayed myofibrils (**K**,**L**,**N2**,**Q**,**R**,**V**), actin blobs which are commonly known as zebra bodies (**M**,**N**,**O**,**R**,**S**,**U**–**AA**), thick or thin myofibrils (**K**,**L**,**O2**,**Q**,**R**,**S**,**U**,**V**,**X**), as well as sarcomeres that are disorganized, short (**N1**,**N2**,**O2**,**Q**,**S**,**T**,**U**) or degenerate (**K**,**N**,**Q**,**V**). Scale bars = 5 μm; phalloidin stained F-actin, white.

**Figure 4 cells-10-02505-f004:**
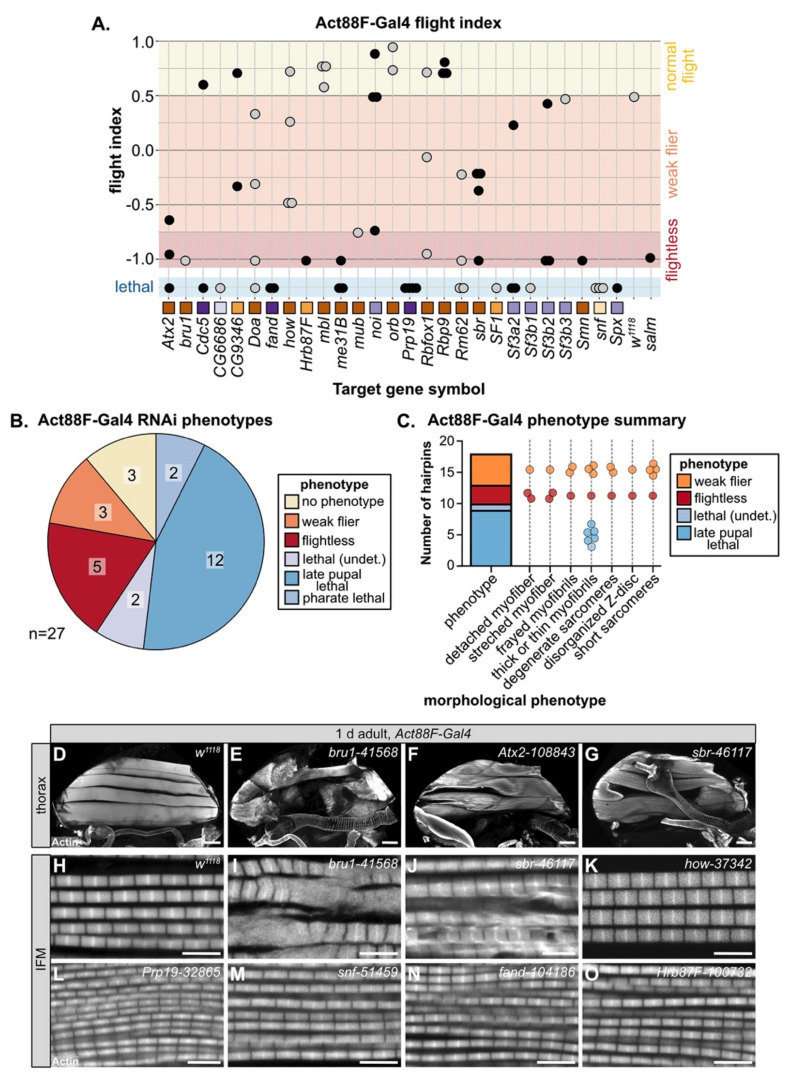
Temporally restricted knockdown of RBPs in IFM with Act88F-Gal4 generates assorted myofibril phenotypes. (**A**) Plot summarizing Act88F-Gal4 phenotype for 65 hairpins targeting 27 RBP genes. Dots represent individual hairpins organized into columns by gene, as labeled. Squares indicate spliceosome membership and are colored as in Figure 2A. Plot background color marks normal flight (yellow), weak/impaired flight (orange), flightless (red) and lethal (blue) phenotypes. (**B**) Pie chart summary of RBP gene-level (strongest hairpin) phenotype when crossed to Act88F-Gal4, which drives in IFM from 24 h APF. **(C)** Graphical summary of confocal phenotypes from 27 hairpins in (**B**). Dots represent hairpins and are colored by flight (flightless, red; weak flier, orange) and lethality (blue shades) phenotype. Hairpins may appear in more than one morphological phenotype category. (**D**–**G**) Confocal Z-projection images showing myofiber phenotypes in 1 d adult flies, including detached or torn myofibers (yellow arrow heads in **E**,**G**) and stretched (yellow arrows in **F**) myofibers. Scale bars = 100 μm. (**H**–**O**) Single-plane confocal images of 1 d adult IFM myofibril and sarcomere structure. RBP knockdown results in diverse phenotypes, such as frayed (**J**) or thin (**L**–**O**) myofibrils as well as degenerate (**I**) or short (**I**–**K**) sarcomeres. Scale bars = 5 μm; phalloidin stained F-actin, white.

**Figure 5 cells-10-02505-f005:**
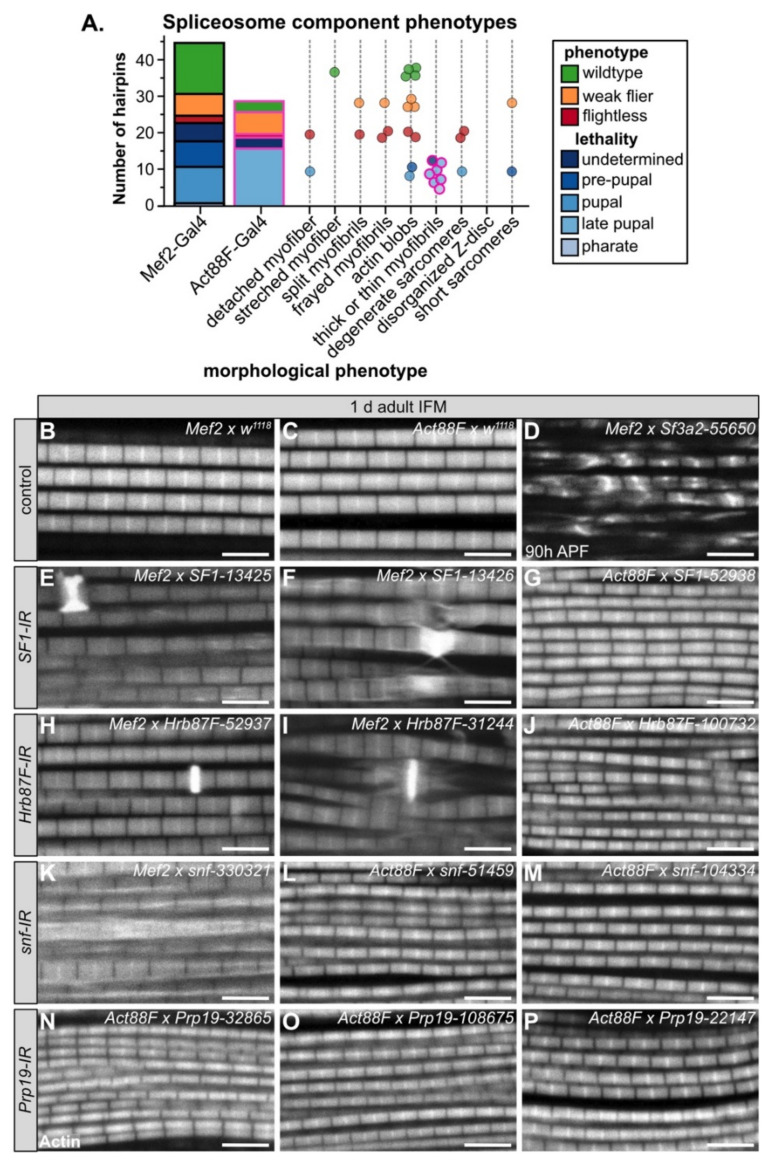
Knockdown of spliceosome core and associated components produces distinctive myofibril and sarcomere phenotypes. (**A**) Graphical summary of lethality, flight and confocal phenotypes for Mef2-Gal4 (black outline) and Act88F-Gal4 (magenta outline) knockdown with 44 hairpins targeting 17 spliceosome genes. Dots represent hairpins and are colored by flight (flightless, red; weak flier, orange) and lethality (blue shades) phenotype. Hairpins may appear in more than one morphological phenotype category. (**B**–**P**) Single-plane confocal images of 1 d adult IFM showing *w^1118^* controls (**B**,**C**) and phenotypes for three hairpins each targeting A-complex components *SF1* (**E**–**G**) and *Hrb87F* (**H**–**J**), A- and B-complex component *snf* (**K**–**M**) or B-, C- and P- complex component *Prp19* (**N**–**P**). A more severe phenotypic example at 90 h APF with knockdown of *Sf3a2* (**D**) is provided for comparison. Mef2-Gal4 knockdown phenotypes are variable, ranging from degenerate (**D,K**), split (**D,I**) and frayed myofibrils (**D,F**) to short sarcomeres (**D,F,I**) and actin blobs (**D**–**F,H**–**I**). Act88F-Gal4 knockdown of spliceosome components results in thin myofibrils (**G, I**–**J,K**–**M,P**). The knockdown efficiency as well as developmental timing influence phenotypic severity. Scale bars = 5 μm; phalloidin stained F-actin, white.

**Figure 6 cells-10-02505-f006:**
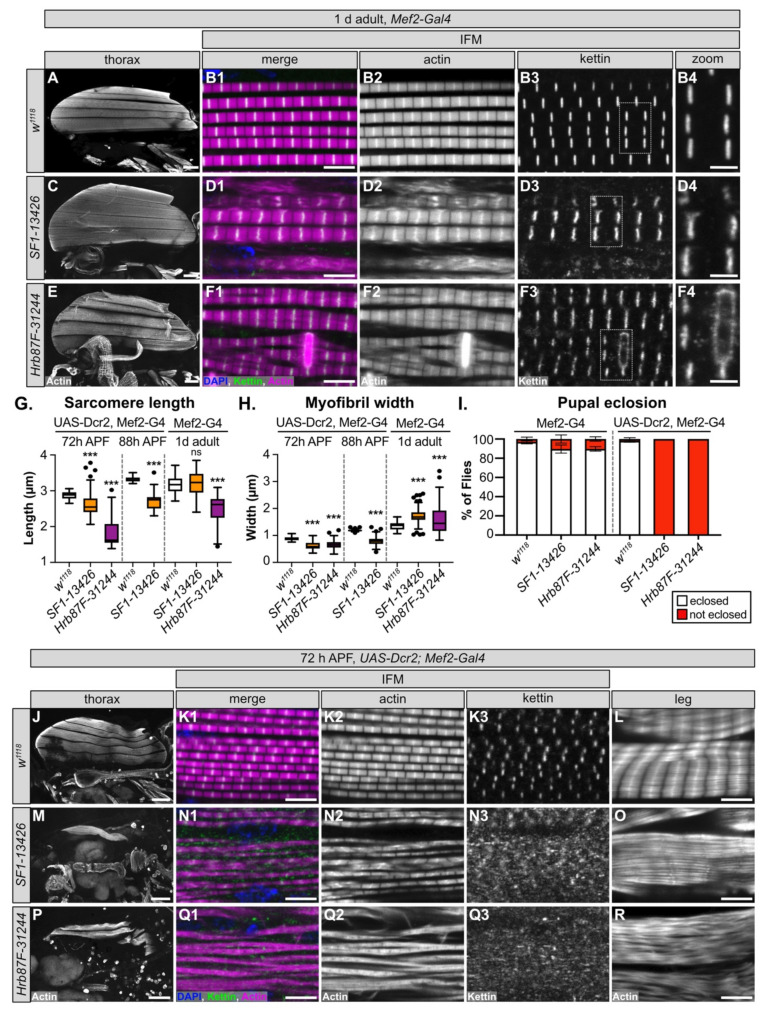
*SF1* and *Hrb87F* are required for proper myofibril development and Z-disc stability. (**A**–**F**) Confocal Z-projection images, table 1. d adults after Mef2-Gal4 driven knockdown of *SF1* (**C**–**D**) and *Hrb87F* (**E**–**F**). Although gross myofiber structure is intact (**A**,**C**,**E**), compared to *w^1118^* controls (**A**–**B**), knockdown myofibrils are thick and split with short sarcomeres and actin-inclusions at the Z-discs (**B2** compared to **D2** and **F2**). kettin staining at the Z-disc is discontinuous and less refined and is aberrant in the vicinity of zebra-bodies (compare **B3,B4** to **D3,D4** and **F3,F4**). Scale bars = 100 μm (**A,C,E**), 5 μm (**B1,D1,F1**) and 2 μm (**B4,D4,F4**). (**G**–**H**) Quantification of sarcomere length (**G**) and myofibril width (**H**) after *SF1* (orange) and *Hrb87F* (purple) knockdown at 72 h APF, 88 h APF or in 1 d adults. Significance is from post hoc Tukey’s test after ANOVA; ***, *p* < 0.001. **(I)** Plot of the percent of pupae with *SF1* or *Hrb87F* knockdown that are able to eclose (white) or remain trapped in the pupal case (red). Genotypes as labeled. Error bars show standard deviation. (**J**–**R**) Confocal Z-projection (**J,L,M,O,P,R**) and single-plane images (**K,N,Q**) showing muscle phenotypes at 72 h APF with UAS-Dcr2, Mef2-Gal4 driven knockdown of *SF1* (**M**–**O**) and *Hrb87F* (**P**–**R**). Enhanced knockdown efficiency results in degraded (**Q2**) and split (**N2**) myofibrils with short sarcomeres. Kettin localization at the Z-disc is impaired and Z-discs are poorly defined (**N1**–**N3,Q1**–**Q3**). Sarcomere structure and organization in tubular leg muscles is also disrupted (**L** compared to **O** and **R**). Scale bars = 100 μm (**J,M,P**), 5 μm (**K1,L,N1,O,Q1,R**).

**Figure 7 cells-10-02505-f007:**
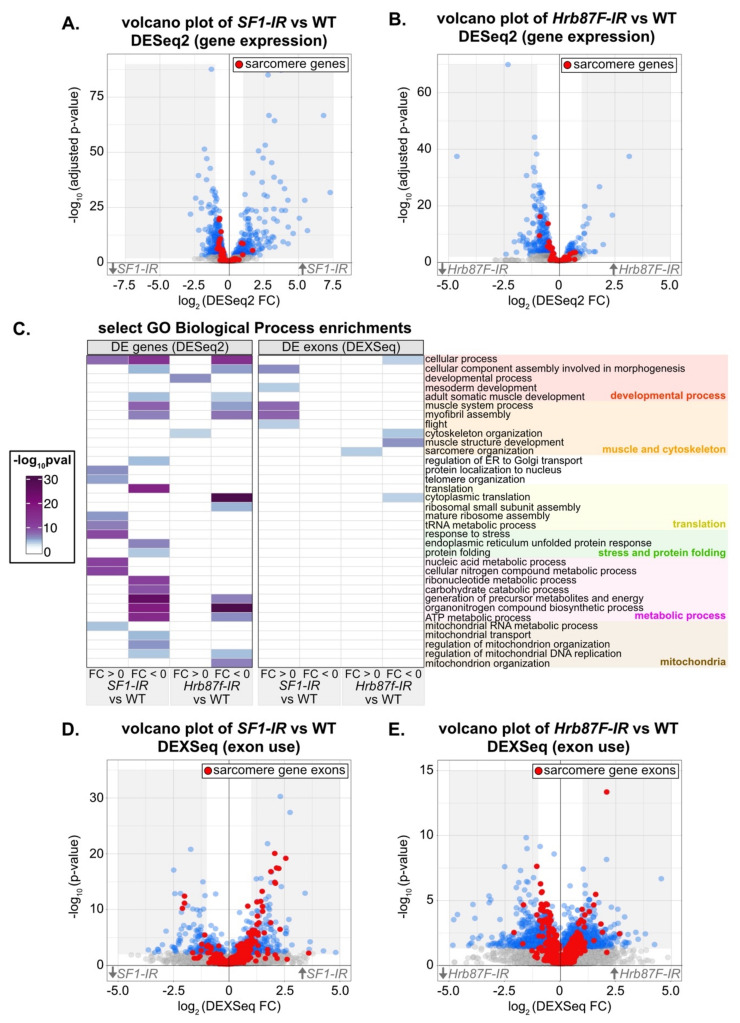
Knockdown of *SF1* and *Hrb87F* alters exon usage of sarcomeric genes. (**A**) Volcano plot of all DESeq2 gene expression data for *SF1-IR* versus WT IFM. Genes with an adjusted *p* value < 0.01 are marked in blue, and genes encoding sarcomere components are labeled red. The gray shaded region marks all genes with a fold change (FC) > 1 or < –1. (**B**) Volcano plot colored as in (**A**) of all DESeq2 gene expression data for *Hrb87F-IR* versus WT IFM. **(C)** Heatmap of select GO Biological Process term enrichments for up and downregulated genes and exons for *SF1-IR* and *Hrb87F-IR* versus WT IFM. A full list of terms can be found in Appendix A. Terms are manually grouped into categories related to a developmental process (red), muscle and cytoskeletal organization (orange), translation (yellow), stress and protein folding (green), a metabolic process (magenta) or mitochondria (brown). (**D**) Volcano plot of DEXSeq changes in exon use for *SF1-IR* versus WT IFM. Exons with a *p* value < 0.01 are marked in blue, and exons from sarcomere genes are labeled red. The gray shaded region marks all exons with a FC > 1 or < –1. (**E**) Volcano plot colored as in (**D**) of DEXSeq exon use data for *Hrb87F-IR* versus WT IFM.

**Figure 8 cells-10-02505-f008:**
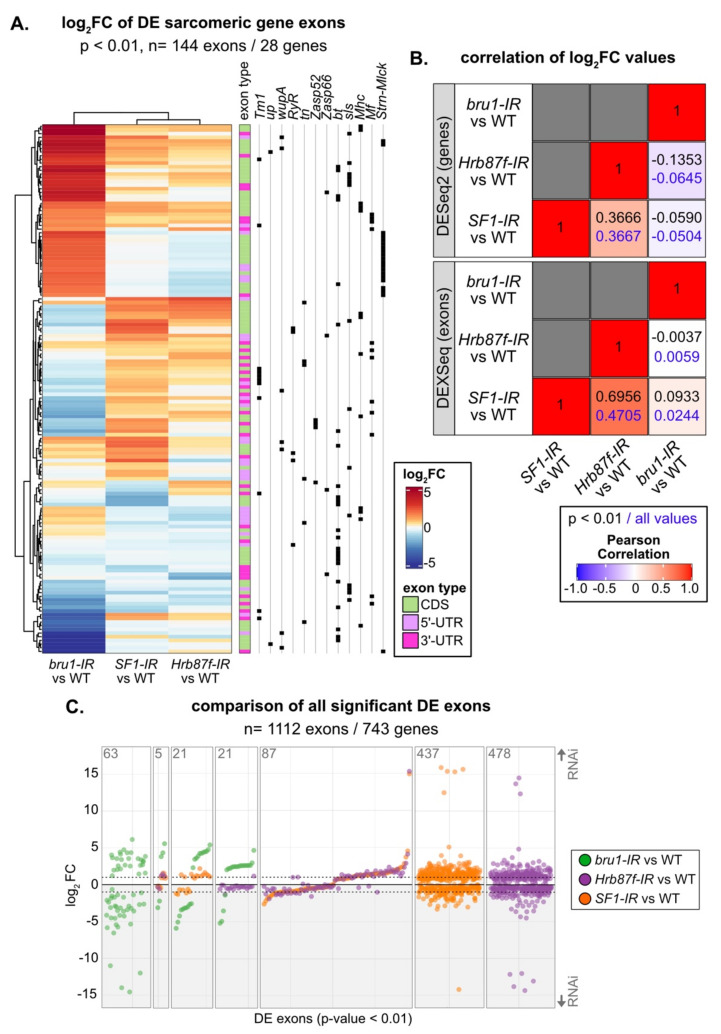
Changes in exon use profiles in *SF1-IR* and *Hrb87F-IR* are distinct from those in *bru1-IR*. (**A**) Heatmap and hierarchical clustering of log_2_FC expression values for all 144 sarcomeric gene exons significantly differentially expressed (DEXSeq *p* value < 0.01) in either *bru1-IR*, *SF1-IR* or *Hrb87F-IR*. Note the similarity in expression between *SF1-IR* and *Hrb87F-IR* and difference to *bru1-IR*, although all three knockdowns affect myofibril and sarcomere development. Exons are classified as coding (CDS, green), 5’-UTR (pink) or 3’-UTR (magenta) based on Flybase annotation. Black squares denote individual exons from the same gene. For multiple sarcomere genes, different sets of exons are up- and downregulated indicating alternative isoform use. (**B**) Plot of Pearson’s correlation coefficients of log_2_FC values from gene expression and exon use analyses between *bru1-IR*, *SF1-IR* and *Hrb87F-IR*. Coefficient values for comparison of all significant genes (DESeq2 adjusted *p* value < 0.01) and exons (DEXSeq *p* value < 0.01) are shown in black, and for comparison of all genes or exons irrespective of significance is shown in blue. (**C**) Plot of log_2_FC values for all 1112 significantly DE exons (DEXSeq *p* value < 0.01) genome wide (*bru-IR* exons, green; *Hrb87F-IR* exons, purple; *SF1-IR* exons, orange). The number of exons is noted in the top left of each plot region. Gray shaded regions contain exons that are downregulated in the knockdown condition. Note the different regulatory dynamics for individual exons between *bru1-IR* and *SF1-IR* or *Hrb87F-IR*, the similar dynamics in regulation between *SF1-IR* or *Hrb87F-IR* and the large number of exons regulated in only one genotype.

## Data Availability

Raw and processed values used to generate plots are available in the Appendix A. mRNA-Seq data are publicly available from GEO with accession numbers GSE184001, GSE63707, GSE107247 and GSE143430.

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
