# Peer review of "A Candidate RNAi Screen Reveals Diverse RNA-Binding Protein Phenotypes in Drosophila Flight Muscle"

_cells, 2021, doi:10.3390/cells10102505_

Round 1

Reviewer 1 Report

Overall Recommendation

  • Accept after Minor Revisions: The paper is in principle accepted after revision based on the reviewer’s comments. Authors are given five days for minor revisions.

Brief summary

The paper entitled “A candidate RNAi screen reveals diverse RNO-binding protein phenotypes in Drosophila flight muscle”. submitted to Cells for review assembles the potential RNA binding proteins from appropriate existing databases and then uses RNAi knockdown to identify new RNA Binding proteins with muscle specific function.  The goal to “provide insight into the regulatory function of individual spliceosome components and demonstrate the efficacy of Drosophila as a genetic model to study muscle-specific RNA regulatory dynamics” is addressed well with appropriate methodology.

Broad comments

In general, the paper is very well organized, well written and well justified.  The paper is written as if experiments were conducted in a strict temporal sequence.  This creates some problems for the reader that are easily addressed.

The introduction assumes the reader has a rather complete understanding of Drosophila muscle types yet fails to mention several well known differences in IFM types.  An introductory paragraph that provides information specific to Drosophila muscle types is needed.  Readership will increase with more complete description of terms and difference in muscle types. This same paragraph should justify the choice of muscle types studied.

Acronyms are often written in full many lines after they are first written in short form.

The results section contains information that could be in methods.  Examples are given below. 

The discussion is excellent.

Specific comments

Line 82.  S2 cells are mentioned first here and again in line 839.  Many papers reference S2 cells with no further explanation, and readers will not necessarily know if these are a subtype or a specific culture. A bit more description of S2 will be appreciated.

Line 160. IFM is not identified as indirect flight muscle until line 231. 

Line 190. APF is undefined.

Muscles are identified as IFM, TDT or leg.  This ignores a great deal that is known about Drosophila muscle types.  IFM for example, are of two distinct types with different developmental pathway.  Salm+ IFM is named in line 277 but not in the Introduction or Methods section.

Reviewer 2 Report

Kao et al. present a detailed and broad bioinformatic and phenotypic analysis of RNA Binding Proteins (RBPs) involved in Drosophila muscle development. The manuscript is well written and needs only minor corrections listed below. The study is exhaustive and gives many informations useful in the muscle field for both wild-type and abnormal development as occurs in human muscle diseases.

Major comments:

  • It needs to be explained how and why the 35 candidate genes studied by RNAi have been selected (line 340).
  • Figure 3B: wild-type seems to have a torn phenotype. Why? if a mistake, this panel needs to be replaced with normal fibres like in panel F.
  • It would be good to show that the IFM phenotype of one or two genetic mutants matches the phenotype of their KO by RNAi.

Minor comments:

  • Line 18: “among” instead of “between” because they are more than two. The same in other sentences of the manuscript.
  • Line 87: comma after “trafficking” instead of “and”.
  • Line 113: “transcriptomic” instead of “transcriptomics” because it is adjective. The same in other sentences.
  • Line 119: delete “to investigate RBP function” (already mentioned above).
  • Line 122: “in the fly” or “in flies”. The same in other sentences.
  • Line 160: IFM has not been mentioned before, so the acronym needs to be explained. I think the system should be described first in the Introduction (at lines 117-122?), and restated at the beginning of Results.
  • Lines 222 and 236: add “the” before “Spliceosome”.
  • Lines 259 and 776: “illustrate” instead of “illustrates”. Data is a plural latin term.
